# Tuning the speed-accuracy trade-off to maximize reward rate in multisensory decision-making

Jan Drugowitsch[1,2,4]*, Gregory C DeAngelis[1†], Dora E Angelaki[3†], Alexandre Pouget[1,4†]

[1]Department of Brain and Cognitive Sciences, University of Rochester, Rochester, United States; [2]Institut National de la Santé et de la Recherche Médicale, École Normale 12 Supérieure, Paris, France; [3]Department of Neuroscience, Baylor College of Medicine, Houston, United States; [4]Département des Neurosciences Fondamentales, Université de Genève, Geneva, Switzerland

**Abstract** For decisions made under time pressure, effective decision making based on uncertain or ambiguous evidence requires efficient accumulation of evidence over time, as well as appropriately balancing speed and accuracy, known as the speed/accuracy trade-off. For simple unimodal stimuli, previous studies have shown that human subjects set their speed/accuracy trade-off to maximize reward rate. We extend this analysis to situations in which information is provided by multiple sensory modalities. Analyzing previously collected data (*Drugowitsch et al., 2014*), we show that human subjects adjust their speed/accuracy trade-off to produce near-optimal reward rates. This trade-off can change rapidly across trials according to the sensory modalities involved, suggesting that it is represented by neural population codes rather than implemented by slow neuronal mechanisms such as gradual changes in synaptic weights. Furthermore, we show that deviations from the optimal speed/accuracy trade-off can be explained by assuming an incomplete gradient-based learning of these trade-offs.

*For correspondence: jdrugo@ gmail.com

†These authors contributed equally to this work

Competing interests: The authors declare that no competing interests exist.

## Introduction

In the uncertain and ambiguous world we inhabit, effective decision making not only requires efficient processing of sensory information, but also evaluating when enough information has been accumulated to commit to a decision. One can make fast, but uninformed and thus inaccurate, decisions or one can elect to make slower, but well-informed, choices. Choosing this so-called speed-accuracy trade-off (SAT) becomes even more complex if several sensory modalities provide decision-related information. For example, the strategy for crossing a busy street will be very different in bright daylight, when one can rely on both eyes and ears to detect oncoming vehicles, as compared to complete darkness, in which case the ears will prove to be the more reliable source of information.

The SAT has been extensively studied for perceptual decisions based on information provided by a single sensory modality. For the most commonly studied visual modality, it has been shown that animals accumulate evidence near-optimally over time (*Kiani and Shadlen, 2009*). In this context, the efficiency of the chosen SAT is assessed in comparison to *diffusion models*, a family of models that trigger decisions as soon as a drifting and diffusing particle reaches one of two bounds (*Ratcliff, 1978*). In these models, which describe the SAT surprisingly well despite their simplicity (*Ratcliff, 1978*; *Palmer et al., 2005*; *Ratcliff and McKoon, 2008*), the drift represents the available sensory information, and the diffusion causes variability in decision times and choices. The level of the bound controls the SAT, with a higher bound leading to slower, more accurate choices. Instructed changes to

the SAT have been shown to be well captured by changes to only the bound in a diffusion model (*Reddi and Carpenter, 2000*; *Reddi et al., 2003*; *Palmer et al., 2005*). Without being explicitly instructed to make either fast or accurate decisions, well-trained human subjects are known to adjust their SAT to maximize their reward rate (*Simen et al., 2009*; *Balci et al., 2011*), or a combination of reward rate and choice accuracy (*Bogacz et al., 2010*). These SAT adjustments are also well captured by tuning the corresponding diffusion model bounds. Thus, we can define the SAT directly in terms of these bounds: a constant SAT refers to behavior predicted by diffusion models with constant bounds, and a SAT that changes across trials requires a diffusion model with bounds that vary on the same time-scale.

Here, we extend the analysis of how human decision-makers adjust their SAT to situations in which they receive information from multiple sensory modalities. We have previously shown that, even in the case of multiple modalities and time-varying evidence reliability, humans are able to accumulate evidence across time and modalities in a statistically near-optimal fashion (*Drugowitsch et al., 2014*). This analysis was based on a variant of diffusion models that retains optimal evidence accumulation even for multiple sources of evidence whose reliability varies differentially over time. As we focused on evidence accumulation in that study, we were agnostic as to how the SAT varied across stimulus conditions; thus, we left the model bounds, which controlled the SAT, as free parameters that were adjusted to best explain the subjects' behavior.

In this follow-up study, we use the previously devised model to analyze whether and how effectively human subjects adjust their SAT if they have evidence from multiple modalities at their disposal. Specifically, we find that subjects adjust their SAT on a trial by trial basis, depending on whether the stimuli are unisensory or multisensory. Moreover, the changes in SAT result in reward rates that are close to those achievable by the best-tuned model, a finding that is robust to changes in assumptions about how the reward rate is computed. Finally, we demonstrate that small deviations from the optimal SAT seem to stem from an incomplete reward rate maximization process. Overall, our findings hint at decision-making strategies that are more flexible than previously assumed, with SATs that are efficiently changed on a trial-by-trial basis.

## Results and discussion

Our analysis is based on previously reported behavioral data from human subjects performing a reaction-time version of a heading discrimination task based on optic flow (visual condition), inertial motion (vestibular condition), or a combination of both cues (combined condition) (*Drugowitsch et al., 2014*). Reliability of the visual cue was varied randomly across trials by changing the motion coherence of the optic flow. Subjects experienced forward translation with a small leftward or rightward deviation, and were instructed to report as quickly and as accurately as possible whether they moved leftward or rightward relative to straight ahead.

First, we ask whether subjects can adjust their SAT from trial to trial. Having related changes in the SAT to changes in diffusion model bounds, this is akin to asking if their behavior could arise from a diffusion model with a bound that changes on a trial-by-trial basis. Our diffusion model necessitates the use of a *scaled* bound, which is the constant *actual* bound per modality divided by the diffusion standard deviation that depends on optic flow coherence. The use of such a scaled bound prohibits us from fitting actual bound levels, but rather scaled versions thereof. For the same reason, we cannot unambiguously predict the behavior that would emerge from a model with actual bounds matched across modalities (i.e., a constant SAT). Therefore, we instead rely on a qualitative argument about how such matched bounds would be reflected in the relation between decision speed and accuracy across modalities.

As *Figure 1A* illustrates for subject B2, increasing the coherence of the optic flow caused subjects to make faster, more accurate choices. This pattern was similar if only the visual modality (solid blue lines, *Figure 1A*) or both modalities were present (solid red lines, *Figure 1A*). This result is qualitatively compatible with the idea that subjects used a single SAT within conditions in which the same modality (visual/vestibular) or modality combination (combined) provided information about heading. Within the framework of diffusion models with fixed actual bounds on the diffusing particle, such a single SAT predicts that, once the amount of evidence per unit time (in our case controlled by the coherence) increases, choices ought to be on average either faster, more accurate, or both in combination, but never slower or less accurate. However, our data violate this prediction, thus showing that the SAT changes across conditions. Consider, for example, the choice accuracy and

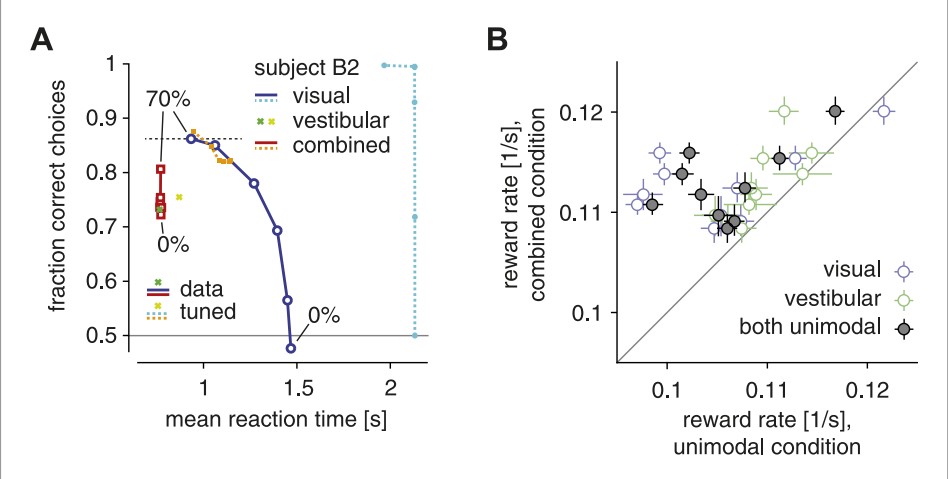

**Figure 1**. The SAT and reward rate for unimodal vs combined conditions. (**A**) Fraction of correct choices is plotted as a function of mean reaction time for subject B2. Blue/cyan: visual condition; green/lime: vestibular condition; red/orange: combined condition. Solid: data; dashed: model with parametric bound tuned to maximize reward rate. Motion coherence varies across data points in the red/orange and blue/cyan curves. The tuned model generally predicts slower and more accurate choices in the visual condition, leading to the longest-possible reaction time (2 s stimulus time + non-decision time) for all but the highest stimulus coherence. (**B**) Reward rate for trials of the combined condition is plotted against reward rate for trials of the visual condition (open blue symbols), the vestibular condition (open green symbols) and both unimodal conditions in combination (gray filled symbols). Reward rates are computed as number of correct decisions per unit time for the respective trial subgroups, and are shown for each subject separately, with bootstrapped 95% confidence intervals.

reaction times of subject B2 in both the visual-only (top blue circle, *Figure 1A*) and combined condition (top red square, *Figure 1A*) trials at 70% motion coherence. Although the combined condition provides more evidence per unit time due to the additional presence of the vestibular modality, responses in the combined condition are less accurate than in the visual-only condition, violating the idea of a single SAT (that is, a fixed diffusion model bound) across conditions. The same pattern emerged across all subjects, whose choices in the combined condition were on average significantly less accurate than in the visual condition (for 70% coherence; one-tailed Wilcoxon signed-rank $W = 54$, $p < 0.002$). As these stimulus conditions were interleaved across trials, our results clearly indicate that subjects were able to change their SAT on a trial-by-trial basis.

A less common variant of diffusion models bounds the posterior belief rather than the diffusing particle. In this case, changing the amount of evidence per unit time only affects the response time but not its accuracy, which remains unchanged. When increasing coherence, we observed a change of both response time and choice accuracy within each condition (*Figure 1A*), supporting a bounded diffusing particle rather than a bounded posterior belief. In rare cases, the two model variants predict the same behavior (*Drugowitsch et al., 2012*), but this is not the case in our context.

Next, we explore whether these adjustments in the SAT serve to maximize subjects' *reward rate*. Even though subjects did not receive an explicit reward for correct trials, we assumed that correct decisions evoke an internal reward of magnitude one. Therefore, we computed reward rate as the fraction of correct decisions across all trials, divided by the average time between the onset of consecutive trials. We proceed in two steps: first, we ask whether subjects have a higher reward rate across trials of the multisensory condition compared to both unimodal conditions. This is an important question because we have found previously that subjects accumulate evidence optimally across modalities (*Drugowitsch et al., 2014*), which implies that, with proper setting of the SAT, they should be able to obtain higher reward rates in the multisensory condition compared to the unimodal conditions. As shown in *Figure 1B*, reward rate is indeed greater, for all subjects, when both sensory modalities are presented than for either modality alone (both unimodal vs combined: Wilcoxon signed-ranks $W = 0$, $p < 0.002$). This confirms that subjects combined evidence across modalities to improve their choices.

We now turn to the question of whether subjects tune their SATs to maximize the reward rate. For this purpose, we focus on the reward rate across all trials rather than for specific stimulus conditions, as subjects might, for example, trade off decision accuracy in unimodal conditions with decision speed in the combined condition. To determine how close subjects were to maximizing their reward rate, we needed to compute the best achievable reward rate. To do this, we tuned the bounds of our modified diffusion model to maximize its reward rate, while keeping all other model parameters, including the non-decision times and choice biases, fixed to those resulting from fits to the behavior of individual subjects. As a starting point, we allowed bounds to vary freely for each stimulus modality and each motion coherence, to provide the greatest degrees of freedom for the maximization. As described further below, we also performed the same analysis with more restrictive assumptions. We call the reward rate resulting from this procedure the *optimal* reward rate. This reward rate was subject-dependent, and was used as a baseline against which the empirical reward rates were compared.

*Figure 2A* shows the outcome of this comparison. As can be seen, all but one subject featured a reward rate that was greater than 90% of the optimum, with two subjects over 95%. As a comparison, the best performance when completely ignoring the stimulus and randomly choosing one option at trial onset (i.e., all actual bounds set to zero) causes a significant 25–30% drop in reward rate (subjects vs random: Wilcoxon signed-rank $W = 55$, $p < 0.002$). Thus, subjects featured near-optimal reward rates that were significantly better than those resulting from rapid, uninformed choices.

Our analysis of the subjects' reward rate relative to the optimum is fairly robust to assumptions we make about how this reward rate and its optimum are defined. Thus far, we have assumed implicit, constant rewards for correct decisions and the absence of any losses for the passage of time or incorrect choices. However, accumulating evidence is effortful, and this effort might offset the eventual gains resulting from correct choices. In fact, previous work suggests that human decision makers incur such a cost, possibly related to mental effort, in the range of 0.1–0.2 units of reward per second for accumulating evidence (*Drugowitsch et al., 2012*). Importantly, this cost modulates both the subjects' and the optimal reward rate, causing the median reward rate across subjects to actually rise slightly to 95.4% and 95.1% (costs of 0.1 and 0.2) of the optimum value (*Figure 2B*, second and third columns), compared to the cost-free median of 93.7%.

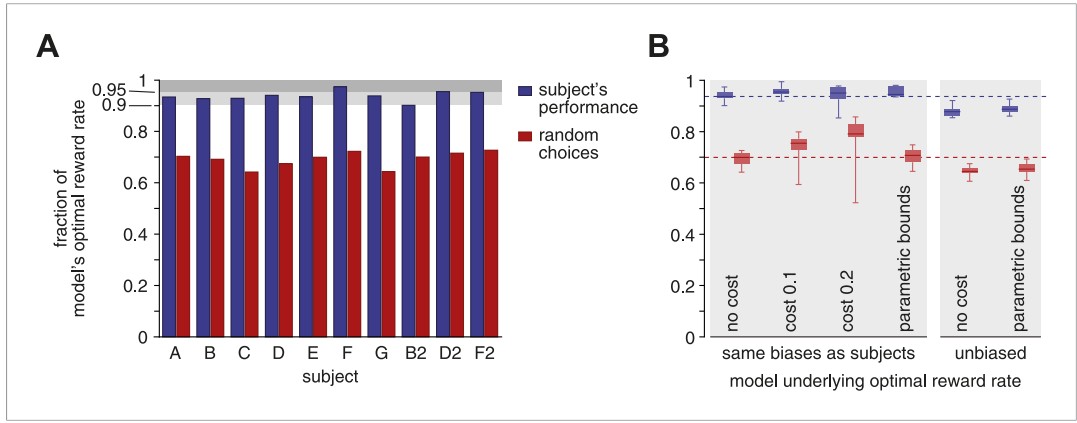

**Figure 2**. Reward rates of subjects relative to the optimal reward rate. The optimal reward rate is the best reward rate achievable by a model with tuned decision bounds. (**A**) Each subject's reward rate is shown as a fraction of the optimal reward rate (blue bars). In addition, the expected reward rate is shown for immediate random decisions (red bars). (**B**) Box-plots show relative reward rates for different assumptions regarding how reward rate is computed. 'no cost' corresponds to the case shown in panel **A**. 'cost 0.1' and 'cost 0.2' assume a cost per second for accumulating evidence over time. 'parametric bounds' uses the original bounds from *Drugowitsch et al. (2014)*, rather than a separate bound parameter for each modality and coherence. The last two bars ('unbiased') remove the subjects' decision biases before computing the optimal reward rate. All box-plots show the maximum/minimum relative reward rates (whiskers), the 25% and 75% percentiles (central bar), and the median (central line) value across subjects. Data are shown for the subjects' reward rates (blue) and for immediate random choices (red).

The optimal reward rates so far were obtained from a model in which we allowed independent bounds for each stimulus modality and each motion coherence, which implies that subjects can rapidly and accurately estimate coherence. Using instead the more realistic assumption (*Drugowitsch et al., 2014*) that bounds only vary across modalities while coherence modulates diffusion variance but not bound height, we reduce the number of parameters and thus degrees of freedom for reward rate maximization. As a result, subjects' reward rates relative to the optimum rise slightly (median 94.5%), where the optimal model is now restricted to use the same bound across all coherences (*Figure 2B*, fourth column). Furthermore, we have assumed the model to feature the same choice biases as the subjects. These biases reduce the probability of performing correct choices, and thus the reward rate, such that removing them from our model boosts the model's optimal reward rate. As a consequence, removing these biases causes a consistent drop in subjects' relative reward rate (*Figure 2B*, last two columns). Even then, reward rates are still around 90% of the optimum (median 87.8% and 88.7% for free and parametric bounds, respectively). If instead of featuring the observed behavior, subjects were to ignore the stimulus and randomly choose one option at trial onset, they would incur a significant drop in reward rate for all of the different assumptions about how we define this optimum (e.g., with/without accumulation cost, …) as outlined above (subject vs random, blue vs red in *Figure 2B*: Wilcoxon signed-rank $W = 55$, $p < 0.002$, except cost 0.2: $W = 54$, $p < 0.004$).

Despite exhibiting near-optimal reward rates, all subjects feature small deviations from optimality. These deviations may result from incomplete learning of the optimal SAT. We only provided feedback about the correctness of choices in early stages of the experiment, until performance stabilized, and subjects did not receive feedback during the main experiment. Nevertheless, subjects' speed/accuracy trade-off remained rather stable after removing feedback, which includes all trials we analyzed. Thus, incomplete learning in the initial training period should be reflected equally in all of these trials. To test the incomplete-learning hypothesis, we assumed that subjects adjusted their strategy in small steps by using gradient-based information about how the reward rate changed in the local neighborhood of the currently chosen bounds. For our argument, it does not matter if the gradient-based strategy was realized through stochastic trial-and-error or more refined approaches involving analytic estimates of the gradient, as long as it involved an unbiased estimate of the gradient. What is important, however, is that such an approach would lead to faster learning along directions of steeper gradients (*Figure 3A*). As a result, incomplete learning should lead to near-optimal bounds along directions having a steep gradient, but large deviations from the optimal bound settings along directions having shallow gradients.

To measure the steepness of the gradient for different near-optimal bounds, we used the reward rate's curvature (that is, its second derivative) with respect to each of these bounds. If these bounds were set by incomplete gradient ascent, we would expect bounds associated with a strong curvature to be near-optimal (red dimension in *Figure 3A*; large curvature, close to optimal bound in inset) and bounds in directions of shallow curvature to be far away from their optimum (blue dimension in *Figure 3A*; small curvature, distant from optimal bound in inset). In contrast, strongly mis-tuned bounds associated with a large curvature (points far away from either axis in *Figure 3B*) would violate this hypothesis. If we plot reward rate curvature against the distance between estimated and optimal bounds, the data clearly show the predicted relationship (*Figure 3B*). Specifically, reward rate curvature is generally moderate to strong in the vestibular-only and combined conditions, and most of these bounds are found to be near-optimal. In contrast, curvature is rather low for the visual condition, and many of the associated bounds are far from their optimal settings. This is exactly the pattern one would expect to observe if deviations from optimality result from a prematurely terminated gradient-based learning strategy. This analysis rests on the assumption that the manner in which reward rate varies with changes in the bounds is well approximated by a quadratic function. If this were the case, then the estimated loss in reward rate featured by the subjects when compared to the tuned model should also be well approximated by this quadratic function. These two losses are indeed close to each other for most subjects (*Figure 3C*), thus validating the assumption.

Previous studies have suggested that deviations from optimal bound settings may arise if subjects are uncertain about the inter-trial interval (*Bogacz et al., 2010*; *Zacksenhouse et al., 2010*). With such uncertainty, subjects should set their bound above that deemed to be optimal when the inter-trial interval is perfectly known. A similar above-optimal bound would arise if subjects are either uncertain about the optimal bound, or have difficulty in maintaining their bounds at the same level across trials. This is because the reward rate drops off more quickly below than above the optimal bounds

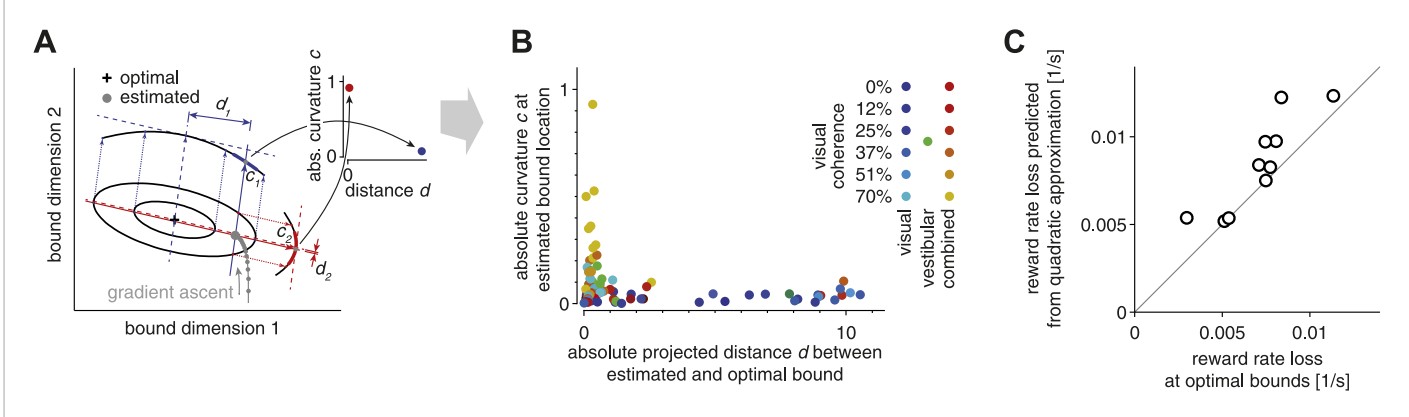

**Figure 3.** Evidence for bound mistuning due to incomplete gradient-based learning. (**A**) The effects of incomplete gradient ascent on the relation between projected bound distance and local curvature (that is, second derivative of the reward rate at estimated bound) are illustrated for a fictional maximization problem with only two bounds. The grey trajectory shows a sequence of gradient ascent steps on the reward rate function, whose shape is illustrated by two iso-reward rate contours (black) around its maximum (cross). Stopping this gradient ascent procedure (large grey filled circle) before it reaches the optimum causes this stopping point to be close to the optimal bound in directions of large curvature (red), and farther away from the optimum in directions of shallow curvature (blue). (**B**) Curvature at the estimated bound location is plotted against the distance between the estimated and optimal bound (see text for details). This plot includes 7 (3 coherence condition) or 13 (6 coherence condition) data points per subject, one for each modality/coherence combination. Data for the visual, vestibular and combined conditions are shown in shades of blue/cyan, green, and red/yellow, respectively, and motion coherence is indicated by color saturation. (**C**) The reward rate loss (i.e., optimal model reward rate minus subject's reward rate) as estimated from the model (abscissa) is plotted against the loss predicted by the quadratic approximation used in the analysis in (**A**)–(**B**), for each subject (ordinate). If the reward rate has a quadratic dependence on the bounds, then all the data points would lie along the diagonal. Small deviations from the diagonal indicate that the reward rate is indeed close-to-quadratic in these bounds.

(*Figure 4A*). Thus, if the subject's bounds fluctuate across trials, or the subjects are uncertain about the optimal bounds, they should aim at setting their bounds above rather than below this optimum. Indeed, this would minimize the probability that the bound would fluctuate well below the optimal value, which would result in a very sharp drop in reward rate. However, our data indicate that, in contrast to previous findings from single-modality tasks (*Simen et al., 2009*; *Bogacz et al., 2010*), subjects consistently set their bounds below the optimum level (*Figure 4B*). In other words, they make faster and less accurate decisions than predicted by either of the above considerations. *Figure 1A* (data vs tuned) illustrates an extreme case for subject B2, in which the best reward rate is achieved in some conditions by waiting until stimulus offset. While not always as extreme as shown for this subject, a distinct discrepancy between observed and reward rate-maximizing behavior exists for all subjects, and is a reflection of the fact that near-optimal reward rates can be achieved with remarkably different joint tunings of reaction times and choice accuracy.

What are the potential neural correlates of the highly flexible decision bounds and associated SATs that are reflected in the subjects' behavior? One possibility is the observed bound on neural activity (*Roitman and Shadlen, 2002*; *Schall, 2003*; *Churchland et al., 2008*; *Kiani et al., 2008*) in the lateral intraparietal cortex in monkeys, an area that seems to reflect the accumulation of noisy and ambiguous evidence (*Yang and Shadlen, 2007*). It still needs to be clarified if similar mechanisms are involved in our experimental setup, in which we observed modality-dependent trial-by-trial changes in the SAT. In contrast to suggestions from neuroimaging studies (*Green et al., 2012*), such trial-by-trial changes are unlikely to emerge from slow changes in connectivity. A more likely alternative, that is compatible with neurophysiological findings, is a neuronal 'urgency signal' that modulates this trade-off by how quickly it drives decision-related neuronal activity to a common decision threshold (*Hanks et al., 2014*). Although only observed for blocked designs, a similar modality-dependent urgency signal could account for the trial-by-trial SAT changes of our experiment, and qualitatively mimic a change in diffusion model bounds. Currently, our model can only predict changes in scaled decision boundaries, which conflate actual boundary levels with the diffusion standard deviation. It does not predict how the actual bound level changes, which is the quantity that relates to the magnitude of such an urgency signal. In general, quantitatively relating diffusion model parameters to neural activity

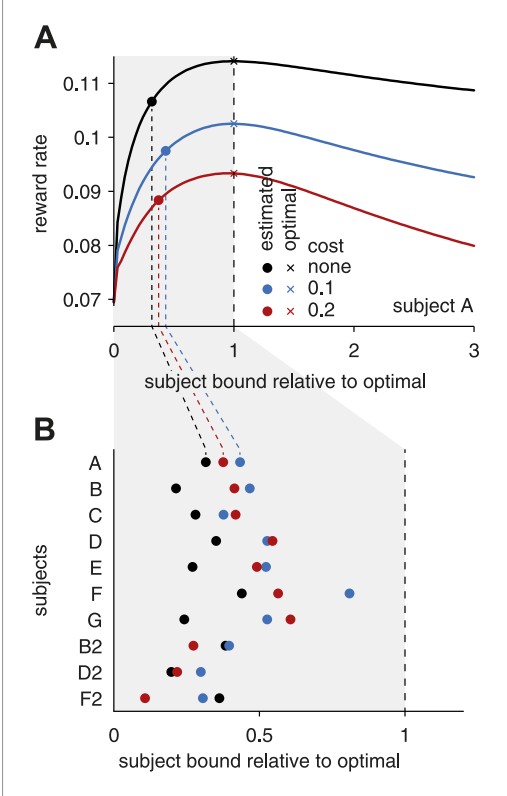

**Figure 4**. Subjects' bound settings relative to the optimal bound. (**A**) The curves show how the reward rate changes with a simultaneous, linear change of all bounds. From left to right, bound levels increase from zero to the (reward-rate maximizing) optimal bound levels (unity values on the abscissa), and continue to bound levels well above this optimum. Different colors correspond to different assumptions about the cost for accumulating evidence over time. The optimal bound levels (unity values on the abscissa) that maximize the reward rate depend on these costs, and thus differ between the three curves. The empirical bound level estimates for individual subjects do not lie on the straight line that is defined by the simultaneous, linear change of all optimal bounds. To evaluate where these empirical bounds lie with respect to the optimal bounds, we found the closest point (along contours of equal reward rate) on this line for the empirical bounds. These points are shown for subject **A** for different costs by the filled circles. (**B**) The closest points are illustrated for all subjects, for different accumulation costs. As can be seen, for any assumption for this cost, the subjects' bounds are well below the optimal settings.

strongly depends on how specific neural populations encode accumulated evidence, which has only been investigated for cases that are substantially simpler (e.g., *Kira et al., 2015*) than the ones we consider here.

Further qualitative evidence for neural mechanisms that support trial-by-trial changes in the SAT comes from monkeys performing a visual search task with different, visually cued, response deadlines (*Heitz and Schall, 2012*). Even though the different deadline conditions were blocked, analysis of FEF neural activity revealed a change in baseline activity that emerged already in the first trial of each consecutive block, hinting at flexible mechanisms that pre-emptively govern changes in SAT. In general, such changes in SAT are likely to emerge through orchestrated changes in multiple neural mechanisms, such as changes in baseline, visual gain, duration of perceptual processing, and the other effects observed by *Heitz and Schall (2012)*, or through combined changes to perceptual processing and motor preparation, as suggested by *Salinas et al. (2014)*.

The observed SATs support the hypothesis that gradient-based information is used by subjects during the initial training trials to try to learn the optimal bound settings. We do not make strong assumptions about exactly how this training information is used, and even a very simple strategy of occasional bound adjustments in the light of positive or negative feedback is, in fact, gradient-based (albeit not very efficient) (e.g., *Myung and Busemeyer, 1989*). The clearest example of a strategy that is not gradient-based is one that does not at all adjust the SAT, or one that does so randomly, without regard to the error feedback that was given to subjects during the initial training period. Such strategies are not guaranteed to lead to the consistent curvature/bound distance relationship observed in *Figure 3B*. For a single speed/accuracy trade-off, adjusting this trade-off has already been thoroughly investigated, albeit with conflicting results (*Myung and Busemeyer, 1989*; *Simen et al., 2006*; *Simen et al., 2009*; *Balci et al., 2011*). Greater insight into the dynamics of learning this trade-off will require further experiments that keep the task stable throughout acquisition of the strategy, and reduce the number of conditions and potential confounds to explain the observed changes in behavior.

In summary, we have shown that subjects performing a multisensory reaction-time task tune their SAT to achieve reward rates close to those achievable by the best-tuned model. This near-optimal performance is invariant under various assumptions about how the reward rate is computed, and is, even under the most conservative assumptions, in the range of 90% of the optimal reward rate. Deviations from optimality are unlikely to have emerged from a strategy of setting bounds to make

them robust to perturbations. Instead, our data support the idea that decision bounds have been tuned by a gradient-based strategy. Such tuning is also in line with the observation of near-optimal reward rates, which are unlikely to result from a random bound-setting strategy. Overall, our study provides novel insights into the flexibility with which human decision makers choose between speed and accuracy of their choices.

## Materials and methods

Seven subjects (3 male) aged 23–38 years participated in a reaction-time version of a heading discrimination task with three different coherence levels of the visual stimulus. Of these subjects, three (subjects B, D, F; 1 male) participated in a follow-up experiment with six coherence levels. The six-coherence version of their data is referred to as B2, D2, and F2. More details about the subjects and the task can be found in *Drugowitsch et al. (2014)*. Not discussed in this reference is the inter-trial interval, which is the time from decision to stimulus onset in the next trial. This interval is required to compute the reward rate, and was 6 s on average across trials.

Unless otherwise noted, we used a variant of the modified diffusion model described in *Drugowitsch et al. (2014)* to fit the subjects' behavior, and we tuned its parameters to maximize reward rates. Rather than using a constant decision bound for each modality and parameterizing how the diffusion variance depends on the coherence of visual motion (as in *Drugowitsch et al., 2014*), the model variant used here allowed for a separate bound/variance combination per modality and coherence. Thus, it featured 7 bound parameters for the 3-coherence experiments, and 13 bound parameters for the 6-coherence experiments. This variant was chosen to increase the model's flexibility when maximizing its reward rate. The original model variant with constant bounds and a changing variance led to qualitatively comparable results (*Figure 1A*, 'tuned', and *Figure 2B*, 'parametric bounds').

For each subject, we adjusted the model's parameters to fit the subject's behavior as in *Drugowitsch et al. (2014)*, through a combination of posterior sampling and gradient ascent. Based on these maximum-likelihood parameters, we then found the model parameters that maximized reward rate by adjusting the bound/variance parameters using gradient ascent on the reward rate, while keeping all other model parameters fixed. To avoid getting trapped in local maxima, we performed this maximization 50 times with random re-starts, and chose the parameters that led to the overall highest reward rate. When performing the maximization, we only modified the parameters controlling the bounds, while keeping all other parameters fixed to the maximum-likelihood values. The latter differed across subjects, such that this maximization led to different maximum reward rates for different subjects. For the 'no bias' variant in *Figure 2B*, we set the choice biases to zero before performing the reward rate maximization.

In all cases, the reward rate was computed as the fraction of correct choices across trials, divided by the average trial time, which is the time between the onsets of consecutive trials. Any non-zero evidence accumulation cost (*Figures 2, 4*) was first multiplied with the average decision time (that is, reaction time minus estimated non-decision time) across all trials, and then subtracted from the numerator.

Our argument about the speed of convergence of steepest gradient ascent is based on the assumption that bounds are updated according to $\theta^n = \theta^{n-1} + \alpha \nabla f(\theta^{n-1})$, where $\theta^{n-1}$ and $\theta^n$ are the bound vectors before and during the $n$th steepest gradient ascent step, $f(\theta)$ and $\nabla f(\theta)$ are the reward rate and its gradient for bounds $\theta$, and $\alpha$ is the step size. The speed of this procedure (i.e., the bound change between consecutive steps) depends for each bound on the size of the corresponding element in the reward rate gradient. For optimal bounds, this gradient is zero, which makes the gradient itself unsuitable as a measure of gradient ascent speed. Instead, we use the rate of change of this gradient close to the bounds $\hat{\theta}$ estimated for individual subjects. This rate of change, called the *curvature*, is proportional to the gradient close to $\hat{\theta}$, and therefore also proportional to the speed at which $\hat{\theta}$ is approached. Close-to and at the optimal reward rate, which is a maximum, this curvature is negative. As we were more interested in its size than its sign, *Figure 3* shows the absolute value of this curvature. We estimated this curvature at $\hat{\theta}$ by computing the Hessian of $f(\hat{\theta})$ by finite differences (D'Errico, John [2006]. Adaptive Robust Numerical Differentiation. MATLAB Central File Exchange. Retrieved 3 July 2014), where we used the model that allowed for a different bound level per modality and coherence (7 and 13 bound parameters/dimensions for 3 and 6 coherence experiment, respectively). Before computing the distance between estimated and reward rate-maximizing bounds, we projected bound parameter vectors into the eigenspace of this Hessian, corresponding to the orientations of decreasing curvature strength. The absolute bound difference was then computed for

each dimension (i.e., modality and coherence) of this eigenspace separately, with the corresponding curvature given by the associated eigenvalue (*Figure 3B*).

In *Figure 3B*, each bound dimension (i.e., modality and coherence, see figure legend) is associated with a different color. As described in the previous paragraph, this figure shows bound differences and curvatures not in the space of original bound levels, but rather in a projected space. To illustrate this bound coordinate transformation in the figure colors, we performed the same coordinate transform on the RGB values associated with each dimension, to find the colors associated with the dimensions of the projected space. The projected colors (filled cirles in *Figure 3B* plot) closely match the original ones (*Figure 3B* legend), which reveals that the curvature eigenspace is well aligned to that of the bound parameters. This indicates that the reward rate curvatures associated with each of the bound parameters, that is, each modality/coherence combination, are fairly independent. Due to the close match between projected and original colors, we do not mention the color transformation in the legend of *Figure 3*.

Our analysis is also valid if subjects do not follow the reward rate gradient explicitly. They could, for example, approximate this gradient stochastically on a step-by-step basis. As long as the stochastic approximation is unbiased, our argument still holds. One such stochastic approximation would be to test if a change in a single bound (corresponding to a single trial) improves the noisy estimate of the reward rate, that is, if $f(\theta^n) > f(\theta^{n-1}) + \varepsilon$, where only a single element (i.e., bound) is changed between $\theta^{n-1}$ and $\theta^n$, and $\varepsilon$ is zero-mean symmetric random noise. In this case, larger changes, which are more likely to occur in directions of larger gradient, are more likely accepted. As a result, faster progress is made along steeper directions, which is the basic premise upon which our analysis is based.

To illustrate how the reward rate changed with bound height (*Figure 4*), we assumed that all (7 or 13) bound parameters varied along a straight line drawn from the origin to the reward rate-maximizing parameter settings. To project the maximum-likelihood bound parameters from the subject fits onto this line (dots in *Figure 4A,B*), we followed the iso-reward rate contour from these parameters until they intersected with the line. We also tried an alternative approach by projecting these parameters onto the line by vector projection, which resulted in a change of the reward rate, but otherwise led to qualitatively similar results as those shown in *Figure 4B*. In both cases, the subjects' bound parameters were well below those found to maximize the reward rate.

## Acknowledgements

Experiments and DEA were supported by NIH grant R01 DC007620. GCD was supported by NIH grant R01 EY016178. AP was supported by grants from the National Science Foundation (BCS0446730), a Multidisciplinary University Research Initiative (N00014-07-1-0937), the Air Force Office of Scientific Research (FA9550-10-1-0336), and the James McDonnell Foundation.

## Additional information

### Funding

| Funder | Grant reference | Author |
|---|---|---|
| National Institutes of Health (NIH) | R01 DC007620 | Dora E Angelaki |
| National Institutes of Health (NIH) | R01 EY016178 | Gregory C DeAngelis |
| National Science Foundation (NSF) | BCS0446730 | Alexandre Pouget |
| U.S. Department of Defense | Multidisciplinary University Research Initiative (N00014-07-1-0937) | Alexandre Pouget |
| Air Force Office of Scientific Research | FA9550-10-1-0336 | Alexandre Pouget |
| James S. McDonnell Foundation | | Alexandre Pouget |

The funders had no role in study design, data collection and interpretation, or the decision to submit the work for publication.

Author contributions

JD, Conception and design, Analysis and interpretation of data, Drafting or revising the article; GCD, AP, Conception and design, Drafting or revising the article; DEA, Conception and design, Acquisition of data, Drafting or revising the article

Ethics

Human subjects: Human subjects: Informed consent was obtained from all participants and all procedures were reviewed and approved by the Washington University Office of Human Research Protections (OHRP), Institutional Review Board (IRB; IRB ID# 201109183). Consent to publish was not obtained in writing, as it was not required by the IRB, but all subjects were recruited for this purpose and approved verbally. Of the initial seven subjects, three participated in a follow-up experiment roughly 2 years after the initial data collection. Procedures for the follow-up experiment were approved by the Institutional Review Board for Human Subject Research for Baylor College of Medicine and Affiliated Hospitals (BCM IRB, ID# H-29411) and informed consent and consent to publish was given again by all three subjects.

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
