## [Decision Letter]

Thank you for sending your work entitled “Tuning the speed-accuracy trade-off to maximize reward rate in multisensory decision-making” for consideration at *eLife*. Your article has been favorably evaluated by Timothy Behrens (Senior editor) and two reviewers, one of whom, Bruno Averbeck, has agreed to share his identity.

The editor and the reviewers discussed their comments before we reached this decision, and agree that the study potentially represents an important advance on your previous work. 

For example:

“This manuscript is essentially a continuation of the work that these authors published recently in *eLife* on multisensory decision making, and as such is an appropriate update/complement to those results.

The current study presents model fits to the original psychophysical data specifically investigating whether the behavior of the subjects was consistent with a strategy to maximize the reward rate. This is indeed an important point because the answer is not entirely obvious given the original observations: in a reaction time (RT) paradigm, when subjects based their perceptual choices on information from two sensory modalities (visual and vestibular) rather than a single one, they generally chose more rapidly rather than more accurately. What the authors found was: (1) that the subjects' behavior across experimental conditions (difficulty x modality) was indeed consistent with reward-rate maximization, and (2) that the observed deviations from optimality were small and can be explained as the consequence of incomplete or imperfect learning.”

However, there are some key points that the reviewers would like addressed before we can reach a final decision.

Most critically, there are: (a) questions about how the model relate to the experimental data:

*Reviewer 1*:

The demonstration that the reward rate is higher in the multimodal than the single-modality conditions is given in Figure 1, and the demonstration that the SAT setting affects the reward rate is given in Figure 2, and there are two important issues about how these key results relate to each other.

In Figure 2 the baseline condition or null hypothesis is the set of red bars, which represent an extreme situation in which decisions are made randomly and as fast as possible; accuracy doesn't matter at all (presumably this corresponds to a zero, or very low bound). It is fine to show that, but the question is whether the SAT is tuned, not whether it exists at all. The relevant comparison is the condition in which the bound is not zero but just constant across conditions; i.e., when the SAT setting is simply fixed. The key piece of information is, how large is the discrepancy between the maximum reward rate obtainable with a variable SAT (i.e., adjustable bound) and one obtainable with a constant SAT (non-adjustable bound)?

A second, related issue, is how those model results square with the experimental data. In particular, suppose you take the best-fitting model with fixed bounds (constant SAT) and make a plot of the model results identical to that in Figure 1. What will it look like? If it matches the real data, then no SAT tuning is necessary across conditions, and conversely, if it does not match the real data then it will provide compelling evidence that the SAT setting does vary across conditions. Then, assuming that the fixed bound is indeed insufficient (which is difficult to infer from the current results), a similar plots can be constructed for the best-fitting model with adjustable bounds. This would directly answer the question of whether the SAT setting is likely to be tuned based on the experimental data, regardless of whether the setting is the optimal one or not. This distinction would be very useful.

*Reviewer 2*:

1) What would the fraction correct look like in Figure 1 for the bound that optimizes reward rate?

2) The SAT is explicitly linked to the bound height in the Introduction and then referred to as SAT throughout. But this was slightly confusing. If one explicitly unpacks the speed-accuracy trade-off then it's not clear what it means to use one for all conditions. Perhaps at times it would be useful to link it back to the bound, or to just say bound when talking about the SAT across conditions. This is particularly confusing because in the final paragraph, the paragraph starts by talking about adjusting the SAT from trial-to-trial and later suggests that one SAT is used across conditions. If this is discussed in terms of the bound it makes sense, but discussing it in terms of the SAT is hard to follow. Also, during the modeling the bounds are allowed to vary by condition. So why is it suggested that a single bound is used? How much do the bounds vary by condition? How close to optimal do the subjects get if a single bound is used across modalities?

and (b) questions about potential circularities in the analysis:

*Reviewer 2*:

The gradient analysis seems circular. Or rather it seems like the results of the gradient analysis are consistent with the subjects doing relatively well. If they set parameters far from optimal in dimensions where the gradient has high curvature they would be quite suboptimal. In fact, it should be possible to relate the total deviation of the subject parameters from optimal, normalized by the hessian to the performance of the subject. In other words, take the difference between the optimal bound parameters and the subject's actual bound parameters (call this delta_b) and multiply them by the inverse of the Hessian. Specifically, delta_b * inv(H) * delta_b. Does this predict how well the subjects do?

However, it is also important that you address the following substantive issues:

The SAT is also confusing because it is in diffusion particle space and not in belief space. If the SAT was in belief space, and various assumptions are met (lack of side bias etc.) then accuracy should be on average the same across conditions, and the speed should be the only thing that changes as information increases, correct? This should perhaps be made more clear and developed in a bit more detail.

Isn't there an explicit link between increasing drift rate (i.e. information rate) and whether speed and/or accuracy both increases for a fixed bound?

Is there a significant difference between subject reward rate and random choices for the cost 0.2 condition? These appear to differ by the least amount.

---

## [Author Response]

*Most critically, there are: (a) questions about how the model relate to the experimental data*:

Reviewer 1:

*The demonstration that the reward rate is higher in the multimodal than the single-modality conditions is given in*
Figure 1*, and the demonstration that the SAT setting affects the reward rate is given in*
Figure 2*, and there are two important issues about how these key results relate to each other*.

*In*
Figure 2
*the baseline condition or null hypothesis is the set of red bars, which represent an extreme situation in which decisions are made randomly and as fast as possible; accuracy doesn't matter at all (presumably this corresponds to a zero, or very low bound). It is fine to show that, but the question is whether the SAT is tuned, not whether it exists at all. The relevant comparison is the condition in which the bound is not zero but just constant across conditions; i.e., when the SAT setting is simply fixed. The key piece of information is, how large is the discrepancy between the maximum reward rate obtainable with a variable SAT (i.e., adjustable bound) and one obtainable with a constant SAT (non-adjustable bound)*?

*A second, related issue, is how those model results square with the experimental data. In particular, suppose you take the best-fitting model with fixed bounds (constant SAT) and make a plot of the model results identical to that in*
Figure 1*. What will it look like? If it matches the real data, then no SAT tuning is necessary across conditions, and conversely, if it does not match the real data then it will provide compelling evidence that the SAT setting does vary across conditions. Then, assuming that the fixed bound is indeed insufficient (which is difficult to infer from the current results), a similar plots can be constructed for the best-fitting model with adjustable bounds. This would directly answer the question of whether the SAT setting is likely to be tuned based on the experimental data, regardless of whether the setting is the optimal one or not. This distinction would be very useful*.

We fully agree that it would have been desirable to address the above questions regarding a constant vs. tuned SAT (and respective model bound) across conditions. Unfortunately, when fitting the diffusion model, we never fit the bound per condition, *ϴ*_*vis*_, *ϴ*_*vest*_ , and *ϴ*_*comb*_, directly. Instead, we only ever fitted the fraction of the bound over the diffusion standard deviation per condition, that is, *ϴ*_*vis*_*/ σ*_*vis*_ (*c*), *ϴ*_*vest*_*/ σ*_*vest*_, and *ϴ*_*comb*_*/σ*_*comb*_(c). These fractions are a sufficient measure to predict reaction times and choice probabilities per condition. As a consequence, reaction times and choice probabilities only allow us to infer these fraction, but not the absolute bound magnitudes independent of the diffusion standard deviations.

This has two important consequences. First, the observed behavior (i.e. choices and reaction times) does not allow us to estimate the subjects’ bounds directly, such we cannot tell by how much they differ for the different conditions. Second, fixing these bounds *ϴ*_j_ in the model to be the same across conditions *j* does not sufficiently constrain this model, as we can rescale the diffusion standard deviations *σ*_*j*_ to achieve arbitrary factions *ϴ*_*j*_/*σ*_*j*_ (c). In other words, setting these bounds to the same level across conditions is not sufficient to guarantee that the SAT is the same across conditions.

Furthermore, we cannot even relate the estimated fraction of bound over diffusion standard deviation, *ϴ*_*j*_/*σ*_*j*_ (c), across conditions *j*, as we assume the diffusion standard deviation *σ*_*j*_ (c) to be a function of the coherence of the visual flow field. Therefore, it varies within conditions, which makes it unclear which coherence to pick as a baseline for comparison.

Overall, it becomes unclear which model parameters to fix to achieve a constant SAT, such that we could not find a model-based definition for a “constant SAT”. For this reason we decided to fall back to using qualitative measures, such as faster, less accurate decisions (e.g. Figure 1) as an indicator for a change in the SAT. The same applies for the baseline reward rate, where we used immediate, random choices, which—as the reviewer correctly points out—corresponds to all bounds set to zero.

Reviewer 2:

*1) What would the fraction correct look like in*
Figure 1
*for the bound that optimizes reward rate*?

We have added this information to Figure 1. It illustrates that the optimally tuned bounds cause slower and more accurate decisions than those featured by the subject. This confirms the analysis underlying Figure 4.

*2) The SAT is explicitly linked to the bound height in the Introduction and then referred to as SAT throughout. But this was slightly confusing. If one explicitly unpacks the speed-accuracy trade-off then it's not clear what it means to use one for all conditions. Perhaps at times it would be useful to link it back to the bound, or to just say bound when talking about the SAT across conditions. This is particularly confusing because in the final paragraph, the paragraph starts by talking about adjusting the SAT from trial-to-trial and later suggests that one SAT is used across conditions. If this is discussed in terms of the bound it makes sense, but discussing it in terms of the SAT is hard to follow. Also, during the modeling the bounds are allowed to vary by condition. So why is it suggested that a single bound is used? How much do the bounds vary by condition? How close to optimal do the subjects get if a single bound is used across modalities*?

The relation between SAT and bound was indeed unclear in the previous version of the manuscript. We now make this link more explicit, first at the end of the second paragraph in the Introduction, and then again in the second paragraph of Results and Discussion. There, we had accidentally stated that this SAT does not change across conditions, which was incorrect, and which we have now fixed.

Due to the reasons outlined further above, we could not relate the bound magnitudes across conditions. The same reasons forbid us to ask how well subjects would fare with a single bound across modalities.

*and (b) questions about potential circularities in the analysis*:

Reviewer 2:

*The gradient analysis seems circular. Or rather it seems like the results of the gradient analysis are consistent with the subjects doing relatively well. If they set parameters far from optimal in dimensions where the gradient has high curvature they would be quite suboptimal. In fact, it should be possible to relate the total deviation of the subject parameters from optimal, normalized by the hessian to the performance of the subject. In other words, take the difference between the optimal bound parameters and the subject's actual bound parameters (call this delta_b) and multiply them by the inverse of the Hessian. Specifically, delta_b * inv(H) * delta_b. Does this predict how well the subjects do*?

Regarding circularity, we agree that it seems counterintuitive to observe a bound distance vs. curvature pattern different from the one we show in Figure 3. Deviations from the optimal bound settings in directions of strong curvature cause larger drops in the reward rate than deviations of similar magnitude in directions of weak curvature (illustrated below by a different spread of iso-reward rate contours in different directions). Thus, if subjects feature close-to-optimal reward rates, one would expect the bounds to be close-to-optimal in directions of strong curvature. However, this might not be the case in directions of weak curvature, as bound mis-tunings in these directions do not strongly impact the reward rate. In other words, one would expect the bounds to be further from optimal in directions of weak curvature, as illustrated in the top panel in Figure 5 (assuming two-dimensional bounds, one dot per subject), just to achieve the observed reward rate. However, this result is not as obvious as it may appear. We could equally well take all these dots and move them along the ellipsoidal iso-reward curves (along which the reward rate does not change) until they are aligned along directions of strong curvature (bottom panel in below figure). Thus, for the same close-to-optimal reward rates we could have observed the opposite pattern—closer- to-optimal bounds in directions of weak curvature rather than directions of strong curvature—but we didn’t. Therefore, close-to-optimal reward rates do not necessarily predict the pattern we observe.

Author response image 1.**DOI:**
http://dx.doi.org/10.7554/eLife.06678.008

The second part of the comment seems to suggest a test for how close the reward rate is to a quadratic function. If this reward rate were perfectly quadratic, then the estimated reward rate loss should coincide with that predicted from the quadratic model. We have performed this analysis (see Figure 3), and the results suggest that there is indeed a close match to the quadratic model, thus validating the implicit assumptions about the functional form of the reward rate that underlies our analysis.

*However, it is also important that you address the following substantive issues*:

*The SAT is also confusing because it is in diffusion particle space and not in belief space. If the SAT was in belief space, and various assumptions are met (lack of side bias etc.) then accuracy should be on average the same across conditions, and the speed should be the only thing that changes as information increases, correct? This should perhaps be made more clear and developed in a bit more detail*.

This is indeed correct and a good point that we omitted in the previous version of the manuscript. We now make clear that the bound is on the diffusing particle, and elaborate in a new paragraph in Results and discussion that this is only in rare cases the same as having a bound on the posterior belief.

*Isn't there an explicit link between increasing drift rate (i.e. information rate) and whether speed and/or accuracy both increases for a fixed bound*?

To our knowledge this link only exists explicitly for time-invariant drift rates within individual trials, in which case there are analytic expressions for mean reaction time and choice probability. In our case, the drift rate changes within individual trials (due to time-varying velocity/acceleration), in which case these quantities needed to be determined numerically.

*Is there a significant difference between subject reward rate and random choices for the cost 0.2 condition? These appear to differ by the least amount*.

The difference is still significant. To make this clear, we have added:

“For all of the different assumptions about how we define this optimum as outlined above, if subjects were to randomly choose one option immediately at trial onset instead of featuring the observed behavior, they would incur a significant drop in reward rate (subject vs. random, blue vs. red in Figure 2: Wilcoxon signed-rank *W*=55, p<0.002, except cost 0.2: *W*=54, p<0.004)” to the relevant paragraph in Results and Discussion.